# Particle Size Effect of Integral Carob Flour on Bioaccessibility of Bioactive Compounds during Simulated Gastrointestinal Digestion

**DOI:** 10.3390/foods11091272

**Published:** 2022-04-27

**Authors:** Ana M. Vilas-Boas, María E. Brassesco, Andreia C. Quintino, Margarida C. Vieira, Teresa R. S. Brandão, Cristina L. M. Silva, Miguel Azevedo, Manuela Pintado

**Affiliations:** 1CBQF—Centro de Biotecnologia e Química Fina—Laboratório Associado, Escola Superior de Biotecnologia, Universidade Católica Portuguesa, Rua Diogo Botelho 1327, 4169-005 Porto, Portugal; amvboas@ucp.pt (A.M.V.-B.); mbrassesco@ucp.pt (M.E.B.); tbrandao@ucp.pt (T.R.S.B.); clsilva@ucp.pt (C.L.M.S.); 2Department of Food Engineering, Campus da Penha, High Institute of Engineering, University of Algarve, 8000-139 Faro, Portugal; andreiacosta94@icloud.com; 3MED—Mediterranean Institute for Agriculture, Environment and Development & CHANGE—Global Change and Sustainability Institute, Instituto Superior de Engenharia, Campus da Penha, University of Algarve, 8005-139 Faro, Portugal; mvieira@ualg.pt; 4Decorgel, Rua do Progresso, 363—Lantemil, 4785-647 Trofa, Portugal; miguelazevedo@decorgel.pt

**Keywords:** carob flour, GIT (gastrointestinal tract) digestion, total phenolic content, antioxidant activity

## Abstract

Carob fruit is native to the Mediterranean region and produced mainly in Portugal, Italy, Morocco and Turkey. The production of the carob fruit in Portugal is highly extensive and sustainable. Currently, carob flour (CF) production is mainly achieved after pulp separation, despite it having been demonstrated that the seeds improve the extraction efficiency of bioactive compounds such as polyphenols, promoting human health. This study aimed to produce an integral CF through an innovative process and assess its physicochemical and bioactive properties at different particle sizes throughout simulated gastrointestinal tract (GIT) digestion. The sugar content profile obtained throughout GIT digestion indicated that sucrose, the sugar present at the highest concentration in undigested CF, was digested and broken down into simple sugars, namely glucose and fructose. The total phenolic content (TPC) and antioxidant activity obtained for the ≤100 µm fraction were in accordance and gastric digestion promoted an increase in the TPC value compared to the undigested sample. The >100 µm fractions displayed a distinct profile from the ≤100 µm fraction. This study showed that the particle size affects the sugar, antioxidant and total phenolic content of CFs and also their gastrointestinal tract digestion. The ≤100 µm fraction demonstrated the most suitable profile as a functional food ingredient.

## 1. Introduction

Carob (*Ceratonia siliqua* L.), a native Mediterranean fruit, is obtained from the evergreen carob tree. It is an edible bean/pod, also known as locust bean or carob pod, and it is composed of pulp (90%) and seed (10%) [1,2,3,4,5]. As a well-adapted xerophytic species, carob trees and pods are cultivated worldwide and produced mainly in Portugal (28.83%), Italy (23.11%), Morocco (16.11%) and Turkey (10.39%), with an average global production rate of 136,612.75 tonnes/year [6,7,8,9]. The Algarve region attains the highest carob pod production in Portugal, allowing this species to grow in warm temperatures with low chilling requirements [6]. The carob pod is characterized by its high sugar content and dietary fiber, low protein content and antioxidant potential [2,4,10,11,12,13]. Relevant compounds present in carob pods, such as sugars, namely sucrose, fructose and glucose, grant their properties as a natural sweetener [14,15].

Studies have shown that carob products, containing phenolic compounds with antioxidant capacity, can promote human health and prevent chronic diseases; thus, their nutritional value and health benefits lead to carob-based products ending up in the food market [2,16]. In this way, the quality and quantity of food components and their digestibility can offer essential nutrients and other bioactive compounds to promote health [7,17].

Several studies on carob-based products have demonstrated the presence of phenolic compounds such as flavonoids, phenolic acids, tannins, etc., specifically in the pulp, seeds and germ of the carob fruit [18]. These compounds have several biological properties, including antioxidant, anti-inflammatory, antibacterial, antifungal and antitumoral activities, glycemic control and gastroprotective and gastrointestinal effects [6]. The digestion process of such compounds promotes their release and degradation from the food matrix, especially during the stomach and intestinal stages, due to the presence of specific enzymes and drastic pH changes [6,19].

Furthermore, awareness of the importance of a balanced diet has grown among consumers; however, many people do not consume enough fruits and vegetables [20,21]. Therefore, the development of new plant-based food products has emerged as a good alternative for granting a sense of satiety, nutritional value and health benefits such as antioxidant, antibacterial, anti-inflammatory and antidiabetic potential [20,21,22,23,24]. However, their nutritional and bioactive profiles may be constrained by gastrointestinal digestion, limiting their bioavailability and efficiency. Thus, a simulated gastrointestinal assay may help to better assess the main compounds released from a food matrix. For example, the release of antioxidants from the food can differ significantly from the antioxidants extracted using chemical extraction assays [25]. These compounds’ bioavailability, stability and bioactivity can vary according to the pH value throughout the gastrointestinal tract, and the presence of digestive enzymes and intestinal microbiota, underlining the importance of investigating these compounds under simulated biological conditions [26].

Due to the differences found in the composition and functional properties, carob pulp and seeds are typically used separately to produce many products, such as flour, gum and syrup, which are widely used in the food (as thickener, stabilizer, cocoa mimetic and sweetener), pharmaceutical and cosmetic industries [1,2,25,27]. Carob flour (CF) production is mainly achieved after pulp separation, despite it having been demonstrated that the presence of the seeds improves the extraction efficiency of bioactive compounds such as polyphenols [6]. In this study, an innovative process to obtain an integral carob flour, composed of both seeds and pulp, was studied regarding its use as a functional food ingredient and the physicochemical and bioactive properties of CFs at different particle sizes were determined.

In this sense, to our knowledge, there is limited information on the effect of the mechanical production process on sugar, total phenolic content and antioxidant activity displayed by the different granulometry of CF fractions. Furthermore, the bioactivity of polyphenols depends on their bioaccessibility when metabolized by the human organism—hence the importance of evaluating their stability and absorption throughout the gastrointestinal digestive tract [10,26,28,29].

Ultimately, this study evaluated the sugar, antioxidant and total phenolic content of the different particle sizes of integral CF throughout simulated gastrointestinal digestion (GIT).

## 2. Materials and Methods

### 2.1. Materials

The 2,2′-azino-bis (3-ethylbenzothiazoline-6-sulphonic acid) (ABTS), 2,2-diphenyl-1-picrylhydrazyl (DPPH), 2,2′-azo-bis-(2-methylpropionamidine) dihydrochloride (AAPH), fluorescein disodium salt, 6-hydroxy-2,5,7,8-tetramethylchroman-2-carboxylic acid (Trolox), hydrochloric acid (HCl), sodium hydroxide (NaOH), α-amylase from porcine pancreas (A3176), pepsin from porcine gastric mucosa (P7000), pancreatin from porcine pancreas (P7545), bile bovine (B3883), D-(−)-fructose (F0127), sucrose (S4100), D-(−)-arabinose (A3131), D-(+)-mannose (M8574), D-(+)-xylose (X3877), D-glucose (G5767) and gallic acid monohydrate were purchased from Sigma-Aldrich (St. Louis, MA, USA). The Folin–Ciocalteu reagent and D-galactose (CAS 59-23-4) were purchased from Merck KGaA (Darmstadt, Germany). The anhydrous sodium carbonate (Na_2_CO_3_) was purchased from PanReac AppliChem GmbH (Darmstadt, Germany). Methanol ≥ 99.9% was purchased from Fisher Scientific (Loughborough, UK). The ultrapure water was obtained from the Milli-Q^®^ Advantage system, Merck KGaA (Darmstadt, Germany).

### 2.2. Carob Flour Preparation

The carob pods were obtained from a regional producer in Faro, Portugal. The pods were harvested in September 2020. Initially, the carob pods were washed with hot running water and placed in an oven at 45 °C for 72 h so that they were free of dirt and moisture and to facilitate the grinding process. The milling process is presented in Figure 1. Briefly, the carob pods were crushed in a knife mill without a sieve, followed by an 8000 µm sieve and a 6000 µm sieve. Then, the obtained milled product was transferred to a hammer mill and milled with a sieve of 3000 µm followed by a 1500 µm sieve. The knife milling step was repeated with the same previous parameters. Finally, the integral CF was milled using an ultracentrifugal mill with a 500 µm sieve. The final CF product was stored and further analyzed for its nutritional and physicochemical properties before and after digestion.

### 2.3. Nutritional and Physicochemical Analysis

All procedures followed the recommendations of the Official Methods of Analysis [27], the International Organization for Standardization [28,29] or the Portuguese regulation [30]. The crude protein content was determined using the Kjeldahl method (ISO 20483:2013) (conversion factor: 6.25). The lipid content was obtained according to method 920.39. The crude ash content was estimated by incineration (Norma NP 518:1986). The moisture content was determined following method ISO 24557:2009. Soluble, insoluble and total dietary fiber content were estimated using the enzyme–gravimetric method, according to AOAC method 991.43 (1990), with slight modifications according to [31]. The starch content was determined by a Megazyme assay, according to AOAC Method 996.11. All measurements were done in triplicate and expressed as grams per 100 g dry basis.

### 2.4. Granulometry Analysis

CF’s particle size distribution was determined with an automatic sieve shaker (Retsch AS 200) with circular oscillation using 50 g of flour and a 5 min sifting time and 1.85 amplitude. Sieves, 20 cm in diameter, with mesh sizes of 100 and 250 µm, were used.

### 2.5. Simulated Gastrointestinal Tract Digestion

The different particle fractions underwent a modified INFOGEST protocol [32], and for each GIT phase, duplicate samples were collected for further analysis.

*Oral phase.* The CF was diluted in pre-warmed simulated salivary fluid (SSF) at a determined ratio to achieve a paste-like consistency; different ratios were considered for the three fractions. The amylase prepared in water was added to the digesta (75 U/mL) and the mixtures were incubated for 2 min at 37 °C at 180 rpm. Duplicate samples were collected before initiating the next phase.

*Gastric phase.* To the previous samples, pre-warmed simulated gastric fluid (SGF) was added at a 1:1 (*v*/*v*) ratio. The pepsin prepared in water was added to the digesta (2000 U/mL) and the pH was adjusted to 3. The mixtures were incubated for 2 h at 37 °C at 120 rpm. Because of the low lipid content in the samples, gastric lipase was not added to the final digesta. Duplicate samples were collected before initiating the next phase. *Intestinal phase*. Pre-warmed simulated intestinal fluid (SIF) was added to the previous samples at a 1:1 (*v*/*v*) ratio. The pancreatin and bile prepared in SIF were added to the digesta (100 U/mL and 10 mM, respectively) and the pH was increased to 7. The mixtures were incubated for 2 h at 37 °C at 60 rpm. The resulting samples underwent simulated dialysis using a dialysis membrane with a molecular pore size of 3.5 kDa (Spectra/Pro^®^6, Spectrum Lab, Breda, The Netherlands) overnight at 37 °C and 60 rpm. Retained samples were collected and absorbed samples were freeze-dried for further analysis.

The oral, gastric and intestinal retained collected samples were then centrifuged using a 3 kDa cut-off filter to reject used digestive enzymes.

#### 2.5.1. Sugar Content

The undigested samples were mixed with 20 mL of 80% ethanol, three times, in order to extract the monosaccharides and oligosaccharides. This mixture was placed in an ultrasonic water bath for 10 min and centrifuged at 5000 rpm for 5 min (AOAC 994.13, 2007). [31]

The undigested and digested samples for different particle fractions were filtered through 0.45 μm Chromafil^®^ PET-45/25 polyester membranes (Macherey-Nagel GmbH, Düren, Germany). The samples were evaluated by chromatographic analysis using an HPLC system with WellChrom HPLC Pump K-1001, Smartline Autosampler 3800 and an RI detector K-2301 (Knauer GmbH, Berlin, Germany). The separation was performed using the Aminex HPX-87H column (BioRad, Hercules, CA, USA) operated at 40 °C; mobile phase, 5 mM H_2_SO_4_; flow, 0.6 mL.min^−1^. The calibration curve for sugar quantification was obtained for sucrose, fructose and glucose.

#### 2.5.2. Total Phenolic Content

The undigested samples (1 g) were mixed with 20 mL of methanol 80%, using an ultraturrax at 24,000 rpm for 30 s. Then, they were incubated in an orbital shaker for 1 h at 200 rpm (room temperature). Finally, they were centrifuged at 8000 rpm for 10 min. The supernatant was collected for total phenolic content and antioxidant activity determinations.

*Folin–Ciocalteu assay.* The total phenolic content (TPC) for the undigested and digested CFs was determined by a Folin–Ciocalteu colorimetric method, as described by Coscueta et al. [33], with some modifications. In a 96-well microplate, 30 µL of each digested sample (diluted in methanol) was mixed with 100 µL of Folin–Ciocalteu reagent (20% *v*/*v*) and 100 µL of anhydrous sodium carbonate solution (7.4% *w*/*v*), in this exact order. The control used was methanol. A standard curve was determined using different concentrations of gallic acid (0.025-0.2 mg/mL). The microplate was incubated at 25 °C for 30 min and the absorbance of the resulting blue mixtures was measured at 765 nm, on a Multidetection plate reader (Synergy H1, Agilent, Santa Clara, Utah, USA). For each particle fraction, all measurements were performed in duplicate. TPC values were expressed in mg of gallic acid equivalents (GAE) per mL of sample.

#### 2.5.3. Antioxidant Activity

The undigested samples (1 g) were mixed with 20 mL of methanol 80%, using an ultraturrax at 24,000 rpm for 30 s. Then, they were incubated in an orbital shaker for 1 h at 200 rpm (room temperature). Finally, they were centrifuged at 8000 rpm for 10 min. The supernatant was collected for total phenolic content and antioxidant activity determinations.

*ABTS assay.* The total antioxidant activity for the undigested and digested CFs was determined by the ABTS method, as described by Gonçalves et al. [34], with some modifications. ABTS was dissolved in water at a final concentration of 7 mM. ABTS radical cation ABTS^•+^ was produced by reacting ABTS stock solution with potassium persulfate (Merck) (final concentration of 2.44 mM) and kept in the dark at room temperature (25 ± 2 °C) for 12–16 h before use. Before the analysis, (ABTS^•+^) was filtered using a 0.22 μm filter (Orange Scientific, Braine-l’Alleud, Belgium) and diluted in methanol to an absorbance of 0.700 ± 0.02 at 734 nm. In a 96-well microplate, 20 µL of the digested sample was mixed with 180 µL of ABTS^•+^ working solution. The control used was methanol. A standard curve was determined using different concentrations of Trolox (25–175 µM). The microplate was incubated at 30 °C for 5 min and the absorbance of the resulting mixtures was measured at 734 nm, on a Multidetection plate reader (Synergy H1, Agilent, Santa Clara, UT, USA). For each particle fraction, all measurements were performed in duplicate. The final results were expressed in µmol of Trolox equivalents per mL of sample.

*DPPH assay.* The total antioxidant activity for the undigested and digested CFs was determined by the DPPH method, as described by Brand-Williams et al. [35], with some modifications. DPPH was dissolved in methanol in a final concentration of 600 µM. Before the analysis, DPPH^•^ was diluted in methanol to an absorbance of 0.600 ± 0.100 at 515 nm. In a 96-well microplate, 25 µL of the digested sample was mixed with 175 µL of DPPH^•^ working solution. The control used was methanol. A standard curve was determined using different concentrations of Trolox (25–250 µM). The microplate was incubated at 25 °C for 30 min and the absorbance of the resulting mixtures was measured at 515 nm, on a Multidetection plate reader (Synergy H1, Agilent, Santa Clara, Utah, USA). For each particle fraction, all measurements were performed in duplicate. The final results were expressed in µmol of Trolox equivalents per mL of sample.

*ORAC assay.* The total antioxidant activity for the undigested and digested CFs was determined by the ORAC method, as described by Dávalos et al. [36], with some modifications. Fluorescein disodium salt (MW = 376.27 g/mol) stock solution was prepared in PBS buffer at 75 mM and pH 7.4. The fluorescein work solution was prepared at 116.66 mM. The control and blank used was PBS buffer. A standard curve was determined using different concentrations of Trolox (10–80 µM). The microplated samples with fluorescein were incubated and shaken thoroughly for 10 min at 37 °C. Subsequently, the AAPH solution (12 mM) was rapidly added to the standard, samples and blank wells and the fluorescence was recorded at intervals of 1 min over 90 min. The excitation wavelength was set at 485 nm and the emission wavelength at 528 nm [37] on a Multidetection plate reader (Synergy H1, Agilent, Santa Clara, UT, USA). For each particle fraction, all measurements were performed in duplicate. The final results were expressed in µmol of Trolox equivalents per mL of sample.

### 2.6. Statistical Analysis

Statistical analysis was performed using GraphPad Prism, v. 8.4.0 software (GraphPad Software, San Diego, CA, USA). The homoscedasticity assumption was met; hence, analysis of variance (two-way ANOVA), with a 95% confidence interval, was applied to every dependent parameter, to assess differences between the different particle fractions and GIT phases in each analysis. Tukey’s test was used for means’ multiple comparisons. In all tests performed, the significance level was set to 5%.

## 3. Results and Discussion

### 3.1. Nutritional, Physicochemical and Granulometry Analyses

The integral carob flour (CF) was analyzed for its proximate composition (Table 1). Results showed high content of total carbohydrates and reasonable content of dietary fiber, namely insoluble fiber, which corresponded to ca. 96% of the fiber present in CF.

The proximate composition is in agreement with previous studies conducted on carob flours, processed by milling approaches, in terms of ash, protein, lipids, carbohydrate and starch content [6,20]. When comparing these results to treated carob flours by seed removal, the greatest differences are in carbohydrate and fiber content, being higher for total fiber content in carob pulp [8,38]. As for carob germ flour and carob bean gums, obtained by seed extraction and purification, the results suggest a similar nutritional profile in regard to moisture, ash, protein and lipids but can noticeably depend on the treatment applied, such as heat or acid treatments and water or acid extractions [39,40,41,42]. The very low starch content determined is similar to previously characterized carob flours [43]. These comparisons allow us to understand that the innovative process used in this study for obtaining an integral carob flour is efficient and produces similar nutritional profiles to those obtained for other carob-based products, described in the literature.

The particle size distribution of CF is presented in Table 2. The results show that 70% of the particles are ≤100 µm, which is expected as most of the carob pod is composed of pulp, grinding more easily than the seed portion, possibly most present in the >100 µm fractions. The different size fractions were assessed regarding their digestion throughout the GIT and antioxidant potential to determine the best-suited fraction for incorporation in food products and health benefits.

### 3.2. Effect of GIT Simulation on Different Particle Fractions of CF—Sugar Content

The simple sugars were identified and quantified by HPLC methodology for CF digested samples throughout the GIT. The sugar content is presented in Figure 2, namely sucrose, fructose and glucose. The sucrose content in undigested CF samples is of a higher quantity than the other sugars, which is in agreement with previous studies [44], and decreases and disappears after gastrointestinal digestion, which can be due to the enzymatic cleavage (amylase) and degradation of glycoside bonds, by extreme conditions such as acidity, and the inversion of sucrose into glucose and fructose units [45]. This is supported by the increase in both fructose and glucose from the oral to the gastric phase, for CF ≤ 100 µm. The larger particle fractions (>100 µm) presented less sugar content when compared to ≤100 µm CF; this is possible since the larger particle fractions might be resulting mostly from the seed part of the carob pod, which exhibits less sugar content than that of pulp and is more difficult to mill, leading to larger particles [12,46].

The results obtained for all sugars were affected by the particle size and the GIT phase (*p* < 0.05). The fraction ≤ 100 µm did not exhibit significant differences (*p* > 0.05) throughout digestion for sucrose, in contrast to glucose and fructose, which differed throughout digestion (*p* < 0.05). The fractions > 100 µm did not show significant differences (*p* > 0.05) throughout GIT digestion for sucrose. In the oral and gastric phase, these fractions did not show significant differences for any sugar (*p* > 0.05) but were significantly different in the intestinal phase.

The fructose and glucose results obtained in the initial oral phase were similar to the results obtained for undigested carob pods compared to previous studies; as for sucrose, undigested carob pods present a much higher sucrose content compared to the results obtained in Figure 2 [47,48,49]. In addition, the World Health Organization (WHO) recommends a sugar intake of less than 10% of total food energy intake because this has been found to be excessive in the population, so carob pods are a good food ingredient for consumption [50].

### 3.3. Effect of GIT Simulation on Different Particle Fractions of CF—Total Phenolic Content (TPC) and Antioxidant Activity (AA)

The bioaccessibility of ingested compounds can be analyzed through in vitro gastrointestinal digestion [10]. The total phenol content (TPC) was determined using the Folin–Ciocalteu methodology for different particle fractions of CF throughout the GIT simulation. The TPC increased (Figure 3—Folin) under gastric digestion, which could possibly be due to the release of phenolic compounds from the carob matrix or the increase in the reactivity of these compounds towards the Folin–Ciocalteu reagent [26,51]. The presence of digestive enzymes and acidic pH appears to influence the release of polyphenolic compounds when observing the values obtained for the oral to gastric phase and undigested samples [52,53].

All three antioxidant activity methodologies showed significant differences between particle fractions and the GIT phase (*p* < 0.05). The GIT phases and particle fractions showed an effect on the results for all assays (*p* < 0.05). The initial oral TPC value for the ≤100 µm particle fraction is in agreement with other studies for carob flours since this matrix is in short-term contact with the oral phase and the amylase activity is limited to the very low starch present in the carob matrices, as determined in Table 1 [9,10,15,48]. When following the digestion of this particle fraction throughout the GIT, an increase in TPC values was observed after the gastric and intestinal phases in comparison to the initial oral phase, as gastric and intestinal digestive enzymes and bile salts act on the food matrix to release these phenolic compounds from the phenolic–protein conjugates, approaching the extracted undigested samples [10,54,55]. This increase in TPC, for the ≤100 µm fraction, is in accordance with the increase in antioxidant activity (Figure 3—ABTS, DPPH and ORAC), since the breakage of high-molecular complexes of the carob matrix and partial deterioration of phenolics result in the production of different types of antioxidant compounds, or pH changes could promote the deprotonation of hydroxyl groups present on the phenolic aromatic rings [10,56,57]. As for the >100 µm particle fractions, the TPC and AA results display a different profile from that of the <100 µm particle fraction, possibly because these fractions contain higher content of insoluble fibers, which can be linked by hydrogen bonding, hydrophobic interactions and covalent bonds with phenolic compounds, hindering the release of said compounds from the carob matrix [54]. Frühbauerová et al. [52] obtained similar results in terms of the effect of particle size for carob flours prepared by cryogenic and vibratory grinding, where smaller particles obtained higher AA and TPC values. As for other studies on vegetables and fruits, an increase in AA and TPC was often observed due to the release of bound phenolics facilitated by intestinal digestive enzymes and bile salts [55,56,57]. Meanwhile, in other studies for carob products, a decrease in these values after intestinal digestion was observed, which could be due to differences in the carob matrix and the preparation method used [10,29]. Hence, several studies on the in vitro GIT digestion for carob-based products have been assessed and shown an influence on the AA and the stability of phenolic compounds under such drastic pH changes and digestive enzymatic activity throughout the GIT; however, limited research has been done to study how particle size affects the bioavailability of these compounds.

## 4. Conclusions

Carob-based products containing phenolic compounds exhibit antioxidant capacity and can promote human health, as well as aiding in preventing chronic diseases. In this study, three different particle size fractions were prepared in order to assess the effect of the particle size on sugar and total phenolic content and antioxidant capacity.

The sugar content profile obtained throughout GIT digestion indicated that the highest content of sugar in undigested carob flours is digested and broken down into simple sugars such as glucose and fructose, potentiating CFs as a functional ingredient within healthy food intake recommendations.

The total phenolic content and antioxidant activity obtained for the ≤100 µm fraction were in accordance, and showed that gastric digestion promoted an increase in the TPC value compared to the chemically extracted samples (undigested). The >100 µm fractions displayed a distinct profile from that of ≤100 µm, possibly due to the higher content of insoluble fiber, which hinders the release of these bioactive compounds from the carob matrix.

These results showed that not only does the preparation method of the integral carob flour allow for an interesting nutritional profile similar to that obtained for more common carob products such as carob pulp, but also the particle size can affect the bioavailability, stability and absorption of the bioactive compounds throughout the GIT. Ultimately, the ≤100 µm fraction exhibited the most suitable profile for use as a functional food ingredient.

## Figures and Tables

**Figure 1 foods-11-01272-f001:**
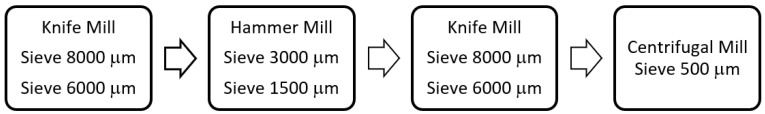
Schematic representation of carob milling process.

**Figure 2 foods-11-01272-f002:**
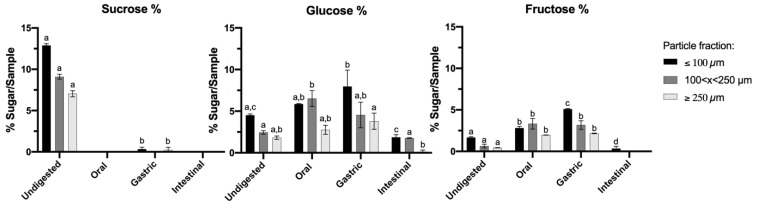
Sucrose, glucose and fructose content in different particle fractions of CF throughout GIT simulation. The bars represent the standard deviation; for a given fraction, results with different letters differ significantly throughout GIT digestion.

**Figure 3 foods-11-01272-f003:**
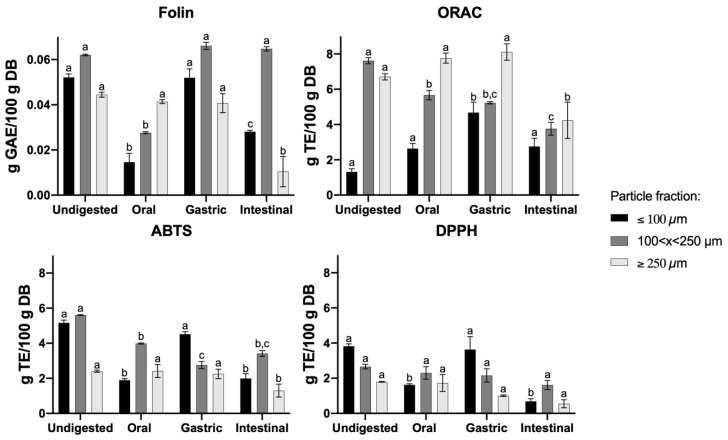
Total phenolic compounds (Folin–Ciocalteu method) and antioxidant capacity (ORAC, ABTS and DPPH assays) of carob flour with different particle sizes in the simulated GIT. All results expressed in g/100 g on dry basis (DB). The bars represent the standard deviation; for a given fraction, results with different letters differ significantly throughout GIT digestion.

**Table 1 foods-11-01272-t001:** Proximate composition of carob flour.

Proximate Composition (g/100 g Dry Basis)	Carob Flour
Moisture	6.35 ± 1.24
Ash	2.41 ± 0.02
Carbohydrate	86.70 ± 1.13
Protein	4.31 ± 0.06
Lipids	0.23 ± 0.12
Energy (kcal/100 g)	366.14 ± 5.60
Total Dietary Fiber	14.64 ± 1.81
Insoluble Dietary Fiber	14.12 ± 1.48
Soluble Dietary Fiber	0.52 ± 0.52
Total Starch	0.96 ± 0.11

**Table 2 foods-11-01272-t002:** Size distribution of the different carob flour fractions.

Particle Size (µm)	Fraction Weight (%)
≤100	70.30 ± 0.31
100 < x < 250	20.18 ± 0.76
≥250	9.52 ± 0.43

## Data Availability

Data is contained within the article.

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
