# Peer review of "Particle Size Effect of Integral Carob Flour on Bioaccessibility of Bioactive Compounds during Simulated Gastrointestinal Digestion"

_foods, 2022, doi:10.3390/foods11091272_

Round 1

Reviewer 1 Report

This work describes the effect of carob flour particle size on the bioaccessibility of bioactive compounds (sugars and phenolic compounds) after the simulated gastrointestinal digestion. The topic of this study is interesting and follows trends in food technology. The introduction is generally well written, however, it should be described more extensively. Authors should provide a better background of the study i.e. information about bioactive compounds and their potential activity. Modes of results presentation are clear. Discussion is supported by results; however, similarly to the introduction, it should be more extensively described – especially the aspect of the stability of phenolic compounds during digestion. Conclusions summarizing the most important findings. In my opinion, the current version of the manuscript needs to be revised and corrected.

Detailed revisions are given below:

Title: consider removing „tract” from the title i.e. „Particle-size effect of integral carob flour on bioaccessibility of bioactive compounds during the simulated gastrointestinal digestion” Please explain the meaning of „integral flour” –it is made from „whole” carob pods (pulp and seeds)?

Line 43-48- To provide a better background of this study describe in more detail the composition (types of compounds) and bioactive properties of carob flour/carob extracts phytochemicals e.g. form in vitro studies. This paragraph should be much more extensively described.

Line 64 - …. integral carob flour (i.e. made of carob pulp and seeds) was studied…?

Line 73 – „Furthermore, the bioactivity of polyphenols”?

Line 103, Figure 1 - The knife milling step was repeated in the same previous parameters. Finally, the integral CF was milled using an ultracentrifugal mill with a 500 μm sieve. – it seems to be contrary to the presented flow chart

 It should be - Knife Mill> Hammer Mill> Centrifugal Mill. Please, check and correct it because the flow chart is unclear.

Line 126 – this subsection should be described in more detail -  amount of flour subjected to digestion, enzyme and bile salts amount and/or activity, simulated salivary fluid composition and ph, usage of gastric lipase etc. If blank samples (without digested material) were prepared for TPC and antioxidant assays? It is known that bile salts and some low molecular components can interfere with the measurements.

Line 127 - The different particle fractions – describe these fractions. Add information about the Particle size range and refer to Table 2

Line 139 - 100 U/mL – based on the alpha-amylase or trypsin activity? Add information.

Line 141-144 - dialysis using a dialysis membrane with a molecular pore size of 3.5 kDa (Spectra/Pro®6, 141 Spectrum Lab, Breda, Netherlands) overnight at 37 °C and 60 rpm. Retained samples were collected and absorbed samples were freeze-dried for further analysis.

Please justify why this additional step was applied > „All collected samples were then centrifuged using a 3 kDa cut-off filter to reject used digestive enzymes.” Why centrifugal filtration was additionally applied after dialysis when, in this case, digestive enzymes were removed after dialysis”

Line 148 – describe the procedure of preparation of undigested samples (similarly for TPC and antioxidant activities)

Line 252-254 – sucrose content „decreases and disappears after gastrointestinal digestion, which can be due to the cleavage and degradation of glycoside bonds and inversion of sucrose into glucose and fructose units [25].”This issue should be more extensively discussed. Please describe which used digestive enzyme or which other factors present during simulated digestion can cleavage and cause degradation of glycoside bonds and inversion of sucrose into glucose and fructose?

Line 257-258 – „this is possible since the bigger particle fractions might be resulting mostly from the seed part of the carob pod, which exhibits less sugar content than that of pulp [12,47].” It is quite a discussive statement regarding the aim of this study (Particle-size effect of integral carob flour on bioaccessibility of bioactive compounds during the simulated gastrointestinal tract digestion). It means that observed effects can be a result of different compositions of obtained fractions instead of only particle size.

Line 279 – the term „bioavailability” should be replaced by the more appropriate term „bioaccessibility”(see cited  INFOGEST procedure and Alminger et al., 2014 - In Vitro Models for Studying Secondary Plant Metabolite Digestion and Bioaccessibility)

Line 282 – „Figure 3 – Folin” >Figure 3 – TPC seems to be more appropriate as the abbreviation of total phenolic content

Line 288-289 – Figure 2 – it should be Figure 3. Move figure described as „Folin” at the top left side because it is described as the first one. Replace “Folin” with “TPC” or “Total phenolic content”

Line 284-287 – consider in the discussion the effect of pH and other digestion factors on the stability of phenolic compounds (line 75)

Line 294-295 – „since this matrix is in short time contact with the oral phase, therefore limiting the amylase activity on this  GIT stage” amylase activity seems to have a little importance in carb flour matrix decomposition because starch represents only about 1% of the matrix (Table 1). Consider it.

Line 338 – “but also the particle size affects the bioavailability, stability and absorption of the bioactive compounds throughout the GIT.” It is only speculation. It wasn't confirmed in this study. Please, rephrase it. E.g. the particle size can affect the bioavailability, stability and absorption of the bioactive compounds throughout the GIT” or similar

Line 301 – add R^2 value

Line 318 – consider the effect of digestion factors on the stability of phenolic compounds in the discussion

Reviewer 2 Report

Type of the Paper (Article

Foods

Article

Particle-size effect of integral carob flour on bioaccessibility of bioactive compounds during the simulated gastrointestinal tract digestion

Authors studied and evaluated sugar, antioxidant, and total phenolic contents of the different particle sizes of integral CF throughout simulated gastrointestinal digestion (GIT). This is an interesting study as there is limited information on the effect of the mechanical production process on sugar, total phenolic content, and antioxidant activity displayed by the different granulometry of CF fractions. Bioactive polyphenols also depend on their bioaccessibility when metabolized by the human organism, hence the importance of evaluating their stability and absorption throughout the gastrointestinal digestive tract. Ultimately, ≤100μm fraction exhibited the most suitable profiling for use as a functional food ingredient.

An adequate number of clear figures and tables were given,

The literature review is also fine, no plagiarism issues

English can be improved,

Accept it with minor changes.
